# W-VSLAM: A Visual Mapping Algorithm for Indoor Inspection Robots

**DOI:** 10.3390/s24175662

**Published:** 2024-08-30

**Authors:** Dingji Luo, Yucan Huang, Xuchao Huang, Mingda Miao, Xueshan Gao

**Affiliations:** 1School of Mechatronical Engineering, Beijing Institute of Technology, Beijing 100081, China; 3120230155@bit.edu.cn (Y.H.); 221068359@stdmail.gxust.edu.cn (X.H.); xueshan.gao@bit.edu.cn (X.G.); 2School of Mechanical Engineering College, Jiangsu Ocean University, Lianyungang 222005, China; 2023000090@jou.edu.cn

**Keywords:** indoor inspection robots, V-SLAM, multi-sensor fusion

## Abstract

In recent years, with the widespread application of indoor inspection robots, high-precision, robust environmental perception has become essential for robotic mapping. Addressing the issues of visual–inertial estimation inaccuracies due to redundant pose degrees of freedom and accelerometer drift during the planar motion of mobile robots in indoor environments, we propose a visual SLAM perception method that integrates wheel odometry information. First, the robot’s body pose is parameterized in SE(2) and the corresponding camera pose is parameterized in SE(3). On this basis, we derive the visual constraint residuals and their Jacobian matrices for reprojection observations using the camera projection model. We employ the concept of pre-integration to derive pose-constraint residuals and their Jacobian matrices and utilize marginalization theory to derive the relative pose residuals and their Jacobians for loop closure constraints. This approach solves the nonlinear optimization problem to obtain the optimal pose and landmark points of the ground-moving robot. A comparison with the ORBSLAM3 algorithm reveals that, in the recorded indoor environment datasets, the proposed algorithm demonstrates significantly higher perception accuracy, with root mean square error (RMSE) improvements of 89.2% in translation and 98.5% in rotation for absolute trajectory error (ATE). The overall trajectory localization accuracy ranges between 5 and 17 cm, validating the effectiveness of the proposed algorithm. These findings can be applied to preliminary mapping for the autonomous navigation of indoor mobile robots and serve as a basis for path planning based on the mapping results.

## 1. Introduction

In recent years, the rapid development of social productivity has led to a significant increase in the demand for security, especially with the continuous expansion of urban areas, which poses greater challenges for building and community security [1]. Additionally, there has been a notable rise in the need for round-the-clock patrol and surveillance in fields such as industrial parks, power facilities, and chemical enterprises. However, the security industry is facing challenges such as an aging workforce and rising labor costs, leading to an urgent need for efficient security robots to meet the demands of environmental perception in complex settings. In this context, visual SLAM technology, a multi-sensor fusion-based localization framework, has gained significant attention [2]. Its application in wheeled robots is particularly notable, as it enables autonomous perception in indoor environments lacking GPS signals, playing a critical role in the development of robot localization and mapping technology.

The most commonly used sensors for SLAM localization are laser and vision sensors. Laser SLAM technology is relatively mature, primarily based on TOF (time-of-flight) LiDAR. For research in laser SLAM, He et al. [3] successfully proposed a laser-inertial pose-estimation and map-construction method using B-spline curves, achieving robust registration steps. In the context of cooperative localization between drones and unmanned vehicles, Ryoga et al. [4] successfully achieved cooperative localization estimation by integrating drone point estimation results with the observation likelihood of unmanned vehicles. Bo et al. [5] combined the IESKF (Iterated Extended Kalman Filter) and factor graphs to propose a more precise local pose estimation method. By filtering LiDAR and IMU data with IESKF, they integrated the laser-inertial odometry with a back-end factor graph, offering new insights into the development of SLAM technology. To address challenges in different scenarios, researchers have proposed SLAM systems combining multi-line LiDAR and visual perception [6,7,8,9], offering diverse solutions for different environments. In the field of firefighting robots, a mapping system based on Cartographer SLAM theory provides autonomous patrol and automatic fire extinguishing functions in unknown indoor environments [10]. However, despite its notable advantages in meeting real-time demands, laser SLAM’s loop closure detection strategies can still present issues with stability, particularly in geometrically symmetric environments. Overall, laser SLAM ensures real-time performance, but its loop closure detection strategies can lag behind, especially in some geometrically symmetric environments where they can lead to unstable loop closures.

The VINS (Visual–Inertial Navigation System) [11] for wheeled robots integrates low-frequency wheel encoder data to enable scale observation. Zhang et al. [12] designed a novel visual-based localization algorithm by combining camera, IMU, and wheel odometer data, providing more comprehensive information for ground robots. Liu et al. [13] tightly fused camera, IMU, and wheel encoder data, proposing a new extended visual–inertial odometry calculation method that significantly improves localization accuracy. In the field of ground vehicles, Lee et al. [14] effectively fused multi-modal visual, inertial, and 2D wheel odometry measurements in a visual–inertial-wheel odometry system using a sliding window filter approach, providing comprehensive information support for ground vehicles. Zheng et al. [15] improved visual estimation accuracy by directly parameterizing the pose of ground vehicles on SE(2) through an efficient pre-integration algorithm based on SE(2). Carlos et al. [16] developed a tightly integrated visual–inertial SLAM system, ORB-SLAM3, that is capable of performing visual, visual–inertial, and multi-map SLAM using monocular, binocular, and RGB-D cameras. Researchers have proposed various innovative methods across different scenarios, such as the planar PnP algorithm by Li J. et al. [17] and Zou’s analysis and comparison of indoor navigation methods based on LiDAR SLAM. Zhu et al. [18] introduced a stereo visual–inertial fusion system based on VINS-Mono that closely integrates visual and inertial measurements. Additionally, Li J. et al. [19] proposed the Attention-SLAM method that combines a visual saliency model with traditional monocular visual SLAM to simulate human navigation patterns. In summary, pure visual SLAM can suffer from unobservable degrees of freedom under certain conditions, leading to estimation errors, particularly for wheeled robots during straight-line motion or constant local acceleration motion. In such cases, the IMU may not provide sufficient information.

In response to the planar movement of wheeled mobile robots in indoor environments, this paper proposes a multi-sensor visual SLAM perception method that integrates visual information with encoder-based wheel odometry information under the SE(2) framework. The high-precision relative pose observations provided by the wheel encoder speed data over short periods allow for accurate inter-frame estimations of visual keyframes. Additionally, the wheel odometry offers reliable pose estimations when visual signals fail (e.g., due to low texture or pure rotation). The integrated visual estimation information also helps to eliminate accumulated pose estimation errors from the wheel encoders, ultimately enabling accurate perception of the pose of the wheeled mobile robot.

## 2. Method Construction

The overall system framework is illustrated in Figure 1. The perception system for the wheeled mobile robot consists primarily of three threads: tracking, local optimization, and global optimization.

The tracking thread runs in real time at a frequency higher than the camera frame rate. Camera information and wheel speed information are soft-synchronized, with the first frame serving as the temporal origin, where the origin of the world coordinate system coincides with the origin of the robot coordinate system, making the first image frame the starting keyframe. First, the wheel speed information with a rate higher than the frame rate is pre-integrated to estimate the initial pose of the mobile robot. Each new image frame undergoes ORB feature extraction, and RANSAC [20,21] is used to remove outliers from feature matches. The thread matches the features in the current frame with the reference keyframe and estimates the initial map points using the pre-integrated information between the feature pairs and image frames. If the current frame meets the criteria for becoming a keyframe, it is added as a new reference keyframe to the local map.

The local optimization thread maintains the local covisibility graph that is generated during the robot’s movement, including keyframes and covisible map points with strong covisibility relationships. When a new keyframe is added to the local map, a search is conducted in the covisibility graph up to three levels deep to find keyframes with a covisibility relationship to the latest keyframe, which are considered local keyframes. Covisible map points associated with each local keyframe are treated as local map points. Local covisibility graph optimization is then performed using the nonlinear optimization library g2o [22] to update the poses of keyframes and the coordinates of map points. After optimization, map points with errors exceeding the threshold are considered outliers and are removed.

The global optimization thread maintains all keyframes and map points within the entire map. Keyframes that have undergone local optimization enter the global optimization thread, where similar keyframes are identified from the entire map using the “bag of words” approach [23] as loop closure candidate frames. Once loop closure is confirmed, the current keyframe, the loop closure keyframe, and their covisible map points form a subgraph. Optimization of the poses of the two keyframes and map points leads to an inter-frame loop closure constraint between the poses of the two frames. Visual constraints, wheel speed pre-integration constraints, and loop closure constraints are then incorporated into the global map to optimize the pose of the mobile robot and global map points, thereby eliminating accumulated errors caused by local optimization.

### 2.1. Coordinate System and Parameter Representation

As shown in Figure 2, the perception problem for wheeled mobile robots primarily involves three coordinate systems: the world coordinate system w, the robot coordinate system b, and the camera coordinate system c. The robot coordinate system can be any frame of reference fixed to the robot body, but in this context, it is defined as the odometry coordinate system. The origin, $O_{1}$, is located at the midpoint of the line connecting the wheel axes, with the positive *x*-axis pointing in the forward direction of the robot. The *y*-axis is aligned with the wheel axis line and positive to the left, while the positive *z*-axis points vertically upward from the origin. The camera coordinate system is centered at the camera’s optical center $O_{2}$, with the positive *z*-axis pointing outwards from the camera lens, and the *x* and *y* axes aligned with the *u* and *v* directions of the pixel plane, respectively.

The Lie groups SO(3) and SE(3) are used to represent rotation *R* and pose *T* in three-dimensional space, respectively:(1)$$\mathrm{SO}\left(3\right)=\left\{R\in R^{3\times 3}\left|RR^{T}=I,\det\left(R\right)=1\right.\right\}$$
(2)$$\mathrm{SE}\left(3\right)=\left\{T=\left[\begin{array}{cc} R & t\\ 0^{T} & 1 \end{array}\right]\in R^{4\times 4}\left|R\in \mathrm{SO}\left(3\right),t\in R^{3}\right.\right\}$$

Since the Lie group space does not support addition, making it impossible to perform small iterations on optimized variables, Lie algebra $\xi ={\left[\phi ,\;r\right]}^{T}\in R^{6}$ is introduced to enable rotations and poses to be mapped as follows: (3)$$\;R=\exp(\phi^{\wedge })\;$$
(4)$$\;T=\exp(\xi^{\wedge })\;$$
In Equation (Equation 3), the Lie algebra $\phi \in R^{3}$ of the rotation matrix corresponds to the rotation vector, which can also be obtained using the Rodrigues formula: (5)$$t=\;J_{l}\left(\phi \right)r$$

The left Jacobian $J_{l}\left(\phi \right)$ is expressed as follows: (6)$$J_{l}\left(\phi \right)=\frac{\sin\left|\phi \right|}{\left|\phi \right|}I+\left(\frac{\left|\phi \right|-\sin\left|\phi \right|}{{\left|\phi \right|}^{3}}\right)\phi \phi^{T}+\frac{1-\cos\left|\phi \right|}{{\left|\phi \right|}^{2}}\phi^{\wedge }$$

Similarly, the pose on SE(2) corresponding to the three-dimensional vector $\varphi ={\left[\nu ,\psi \right]}^{T}$ can be expressed as follows: (7)$$\vartheta =\left[\begin{array}{cc} \Phi & \nu \\ \;0^{T} & 1 \end{array}\right]$$
where
(8)$$\Phi =\exp{\left(\psi \right)}^{\wedge }=\left[\begin{array}{cc} \cos\psi & -\sin\psi \\ \sin\psi & \cos\psi \end{array}\right]$$

The correspondence between the Lie algebra se(3) and se(2) can be expressed as follows: (9)$$\mathrm{se}\left(3\right)\mapsto \mathrm{se}\left(2\right):\varphi ={\left[r_{x},r_{y},\phi_{z}\right]}^{T}$$
(10)$$\mathrm{se}\left(2\right)\mapsto \mathrm{se}\left(3\right):\xi ={\left[\nu_{1},\nu_{2},0,0,0,\psi \right]}^{T}$$

In general, the transformation matrix $T_{\mathrm{AB}}$ represents the transformation from coordinate system *B* to coordinate system *A*. The matrix includes a rotation matrix $R_{\mathrm{AB}}$ and a translation vector $t_{\mathrm{AB}}$. The pose of a ground mobile robot is represented in SE(2) and denoted as $T_{\mathrm{wb}}$, while the pose of the camera is represented in SE(3) and denoted as $T_{\mathrm{cw}}$. The external calibration transformation matrix between the robot and camera coordinate systems is also represented in SE(3) and denoted as $T_{\mathrm{bc}}$. A 3D map point is expressed in the world coordinate system as $P_{w}$, and the map point observed in the camera coordinate system is denoted as $P_{c}$. The transformation relationship between these coordinates is expressed as follows: (11)$$\;P_{c}=T_{\mathrm{cw}}P_{w}\;$$

### 2.2. Visual SLAM Perception Algorithm with Wheel Speed Fusion

Since optimization-based SLAM back-end methods can iteratively achieve more accurate state estimates, the sensing problem of wheeled mobile robots moving on indoor planes is modeled here as a maximum a posteriori estimation problem (MAP). Given the initial pose $T_{wb_{0}}$ of the wheeled mobile robot, the camera intrinsic matrix *K*, and the external calibration matrix $T_{\mathrm{cb}}$, and considering a series of continuous input images $c_{i}$ (*i* = 1, 2, ⋯) and wheeled odometer speed information $\varpi_{j}$ (*j* = 1, 2, ⋯), the goal is to estimate a series of poses $T_{wb_{i}}$ (*i* = 1, 2, ⋯) of the wheeled mobile robot on SE(2) and the corresponding map points $P_{w}^{k}$ (*k* = 1, 2, ⋯) throughout the observation process. Next, we construct visual observation constraints, wheeled pre-integration constraints, and loop closure observation constraints to form the final MAP problem.

### 2.3. Visual Observation Constraint

As shown in Figure 3, the camera uses a pinhole model to map the 3D coordinate point $P_{w}$ in the world coordinate system to the 2D coordinate $P_{uv}\left(u,v\right)$ on the pixel plane.

In the camera coordinate system, the relationship between the 3D point $P_{c}$ and the 2D point $P_{uv}$ on the pixel plane is as follows: (12)$$\left[\begin{array}{c} u\\ v \end{array}\right]=\frac{1}{z}\left[\begin{array}{ccc} f_{x} & 0 & c_{x}\\ 0 & f_{y} & c_{y} \end{array}\right]P_{c}$$

In the formula, *z* is the *z*-axis component of the 3D coordinate point. The parameters $f_{x}$ and $f_{y}$ represent the camera focal lengths along the *x* and *y* axes, respectively. The parameters $c_{x}$ and $c_{y}$ denote the coordinates of the image center point on the pixel plane.

This can be abbreviated as the projection function: (13)$$P_{uv}=\pi \left(P_{c}\right)$$

Given that visual measurements may be affected by noise, under the assumption of a Gaussian model, the measurement equation can be expressed as follows: (14)$$P_{uv}=\pi \left(P_{c}\right)+\eta_{uv}$$
where $\eta_{uv}∼N\left(0,\sigma_{uv}^{2}\right)$.

To facilitate the nonlinear optimization solution of the SLAM backend, we differentiate the previous equation with respect to the 3D point $P_{c}$ in the camera coordinate system to obtain the Jacobian matrix of the pixel observation with respect to the camera coordinate point $P_{c}$ [24]: (15)$$J_{\pi }\left(P_{c}\right)=\frac{1}{z}\left[\begin{array}{ccc} f_{x} & 0 & -x\frac{f_{x}}{z}\\ 0 & f_{y} & -y\frac{f_{y}}{z} \end{array}\right]$$

When the *i*-th keyframe observes a landmark point $P_{w}^{j}$ and the camera extrinsic parameter rotation matrix $R_{\mathrm{cb}}$ and translation vector $t_{\mathrm{cb}}$ are known, the observation model in the camera coordinate system is given by
(16)$$P_{uv}\left(T_{wb_{i}},P_{w}^{j}\right)=\pi \left(R_{\mathrm{cb}}R_{wb_{i}}^{T}\left(P_{w}^{j}-P_{wb_{i}}\right)+t_{\mathrm{cb}}\right)+\eta_{uv}$$

In the equation, $R_{wb_{i}}$ and $P_{wb_{i}}$ represent the SE(3) expressions corresponding to the robot’s pose parameterized on SE(2), while π(·) denotes the camera projection equation.

Unlike the unconstrained state of SE(3), a mobile robot undergoing planar motion is parameterized as a pose on SE(2). Given the minor vibrations that occur during the motion of the robot [15], perturbations in SE(3) are added to the SE(2) pose of the robot to account for the existing visual measurement noise. Assuming the translational perturbation experienced by the robot’s pose is $\eta_{t}∼N\left(0_{3\times 1},\sigma_{t}^{2}\right)$ and the rotational perturbation is $\eta_{\theta }∼N\left(0_{3\times 1},\sigma_{\theta }^{2}\right)$, the actual pose of the robot should be represented as
(17)$$R_{wb_{i}}\leftarrow \exp\left(\eta_{\theta }\right)R_{wb_{i}}\;$$
(18)$$P_{wb_{i}}\leftarrow \;P_{wb_{i}}+\eta_{t}$$

Therefore, the observation equation in Equation (Equation 16) is modified as follows: (19)$$P_{uv}\left(T_{wb_{i}},P_{c}^{j}\right)=\pi \left(R_{\mathrm{cb}}\exp\left(\eta_{\theta }\right)R_{wb_{i}}^{T}\left(P_{c}^{j}-P_{wb_{i}}-\eta_{t}\right)+t_{\mathrm{cb}}\right)+\eta_{uv}$$
From the above Equation (Equation 19), it is evident that the pixel observation is influenced by three types of noise: $\eta_{\theta }$, $\eta_{t}$, and $\eta_{uv}$. To calculate the covariance of the camera observation, a first-order Taylor expansion is performed:(20)$$P_{uv}\left(T_{wb_{i}},P_{c}^{j}\right)\approx \pi \left(T_{wb_{i}},P_{c}^{j}\right)+J_{\eta_{\theta }}\eta_{\theta }+J_{z}\eta_{t}+\eta_{uv}$$
The Jacobian matrix of the camera observation with respect to the rotational perturbation $\eta_{\theta }$ and translational perturbation $\eta_{t}$ is given by
(21)$$J_{\eta_{\theta }}=J^{\pi }\left(T_{wb_{i}},P_{c}^{j}\right)R_{\mathrm{cb}}R_{wb_{i}}^{T}{\left(P_{c}^{j}-P_{wb_{i}}\right)}^{\wedge }\;$$
(22)$$\;J_{z}\eta_{t}=-J^{\pi }\left(\rho_{i},l_{j}\right)R_{\mathrm{cb}}R_{wb_{i}}^{T}\;$$
The term $J^{\pi }$ refers to the Jacobian matrix of the camera projection equation, as shown in Equation (Equation 15).

Thus, we have
(23)$$\;\delta \eta =J_{\eta_{\theta }}\eta_{\theta }+J_{t}\eta_{t}\delta +\eta_{uv}\;$$

Therefore, the covariance propagation equation for camera observation information under perturbation can be derived as follows: (24)$$\Sigma_{ij}=J_{\eta_{\theta }}\Sigma_{\theta }J_{\eta_{\theta }}^{T}+J_{\eta_{t}}\Sigma_{t}J_{\eta_{t}}^{T}+\Sigma_{uv}$$

Subsequently, under the aforementioned constraints, the camera’s reprojection observation error equation can be constructed as follows:(25)$$e_{ij}=\pi \left(T_{wb_{i}},P_{c}^{j}\right)-P_{uv}\left(T_{wb_{i}},P_{c}^{j}\right)$$

The Jacobian matrices of the reprojection observation error with respect to the rotational and translational components of the robot’s pose are, respectively,
(26)$$\frac{\partial e_{ij}}{T_{wb_{i}}}=\left[\begin{array}{cc} \frac{\partial e_{ij}}{\partial \theta_{i}} & \frac{\partial e_{ij}}{\partial \nu_{i}} \end{array}\right]$$
(27)$$\frac{\partial e_{ij}}{\partial \theta_{i}}=-J^{\pi }\left(T_{wb_{i}},\;l_{j}\right)R_{\mathrm{cb}}R_{wb_{i}}^{T}\Lambda_{12}$$
(28)$$\frac{\partial e_{ij}}{\partial \nu_{i}}=J^{\pi }\left(T_{wb_{i}},\;l_{j}\right)R_{\mathrm{cb}}R_{wb_{i}}^{T}{\left(P_{c}^{j}-P_{wb_{i}}\right)}^{\wedge }e_{3}$$

The Jacobian matrix of the reprojection error with respect to the landmark point is: (29)$$\frac{\partial e_{ij}}{\partial P_{c}^{j}}=J^{\pi }\left(T_{wb_{i}},\;P_{c}^{j}\right)R_{\mathrm{cb}}R_{wb_{i}}^{T}$$

At this point, we have derived the camera reprojection observation error, its corresponding covariance, and the Jacobian matrix of the error with respect to the variables to be optimized.

### 2.4. Wheel Speed Pre-Integration Constraint

In wheeled mobile robots, encoder information is typically generated at a higher frequency than visual data, resulting in each frame of visual data corresponding to multiple instances of encoder data. Consequently, updating the camera pose at each iteration of back-end optimization, alongside the robot poses represented by the multiple sets of encoder data, can either lead to redundant computations or affect the accuracy of perception. Therefore, the concept of pre-integration is employed [25], as shown in Figure 4, to construct constraints between frames of visual information using pre-integrated encoder data.

First, the linear velocity *v* and angular velocity ω of a wheeled mobile robot can be derived from the left and right wheel speeds, as well as the robot’s intrinsic parameters.
(30)$$\left[\begin{array}{c} v\\ \omega \end{array}\right]=\frac{1}{2B}\left[\begin{array}{cc} B & B\\ 1 & -1 \end{array}\right]\left[\begin{array}{c} v_{r}\\ v_{l} \end{array}\right]+\;n_{v\omega }$$

In the equations, the linear velocity and angular velocity are subject to Gaussian noise, $n_{vw}∼N\left(0,\sigma_{vw}^{2}\right)$. To mitigate errors from manual measurements, accurate intrinsic parameters of the robot should be obtained through joint calibration with the camera [26].

Then, the discrete integration model for wheel speed at two consecutive time points can be derived, with a sampling interval of *T*.
(31)$$\theta_{k+1}=\theta_{k}+\omega_{k}T$$
(32)$$p_{k+1}=p_{k}+R\left(\theta_{k}\right)\left[\begin{array}{c} 0\\ v_{k}T \end{array}\right]$$
where $R\left(\theta_{k}\right)=\left[\begin{array}{cc} \cos\theta_{k} & -\sin\theta_{k}\\ \sin\theta_{k} & \cos\theta_{k} \end{array}\right]$.

Given $\varpi_{k}={\left[v_{k},\omega_{k}\right]}^{T}$, the above equation can be expressed in a compact form as
(33)$$q_{k+1}=f\left(q_{k},\;\varpi_{k}\right)$$
Performing a Taylor expansion on the equation yields the error state equation as follows: (34)$$\delta q=\frac{\partial f}{\partial q_{k}}q_{k}+\frac{\partial f}{\partial \varpi_{k}}\varpi_{k}$$
(35)$$\frac{\partial f}{\partial q_{k}}=\left[\begin{array}{ccc} 1 & 0 & -v_{k}\sin\theta_{k}T\\ 0 & 1 & v_{k}\cos\theta_{k}T\\ 0 & 0 & 1 \end{array}\right]$$
where $\frac{\partial f}{\partial q_{k}}=\left[\begin{array}{ccc} 1 & 0 & -v_{k}\sin\theta_{k}T\\ 0 & 1 & v_{k}\cos\theta_{k}T\\ 0 & 0 & 1 \end{array}\right]$, $\frac{\partial f}{\partial \varpi_{k}}=\left[\begin{array}{cc} \cos\theta_{k} & 0\\ \sin\theta_{k} & 0\\ 0 & T \end{array}\right]$.

Given $F_{k}=\frac{\partial f}{\partial q_{k}}G_{k}=\frac{\partial f}{\partial \varpi_{k}}$, the error state equation between two consecutive wheel speed moments can be obtained as follows: (36)$$\left[\begin{array}{c} \delta \theta_{k+1}\\ \delta p_{k+1} \end{array}\right]=\;F_{k}\left[\begin{array}{c} \delta \theta_{k}\\ \delta p_{k} \end{array}\right]+\;G_{k}\left[\begin{array}{c} n_{\omega }\\ n_{v} \end{array}\right]$$

In this way, the covariance propagation equation for wheel speed observations at two consecutive time intervals can be derived [27]: (37)$$P_{k+1}=\;F_{k}P_{k}F_{k}^{T}+G_{k}Q_{v\omega }G_{k}^{T}$$

The initial value of $P_{k}$ is generally set as a zero matrix, and $Q_{vw}$ is derived from the measurement noise $n_{vw}$ of the encoder wheel speed: (38)$$Q_{v\omega }=\left[\begin{array}{cc} \sigma_{v}^{2} & 0\\ 0 & \sigma_{\omega }^{2} \end{array}\right]$$

Assuming there are *n* instances of wheel speed information $\varpi_{k+1},\;\varpi_{k+2},\cdots ,\;\varpi_{k+n}$ between two consecutive keyframes $c_{i}$ and $c_{i+1}$, as indicated by Equation (Equation 33), the wheel speed pre-integration observation between the two keyframes is as follows: (39)$$\Delta q_{i,i+1}=\sum_{i=k+1}^{k+n}f\left(q_{i},\varpi_{i}\right)$$

Accordingly, the covariance matrix of the pre-integrated quantities $\Sigma_{i,i+1}$ between two visual keyframes can be recursively obtained from Equation (Equation 37).

Then, let the pre-integrated observation be denoted as ${\bar{T}}_{i,i+1}=\Delta q_{i,i+1}$, and based on the pre-integrated quantities derived in Equation (Equation 39), the wheel speed measurement error equation can be constructed as follows: (40)$$e_{i,i+1}={\bar{T}}_{i,i+1}T_{wb_{i}}T_{wb_{i+1}}^{-1}$$

By dividing the residuals into rotational and translational components, we obtain
(41)$$\left[\begin{array}{c} e_{i,i+1}^{\theta }\\ e_{i,i+1}^{\rho } \end{array}\right]=\left[\begin{array}{c} {\bar{\theta }}_{i,i+1}-\left(\theta_{i+1}-\theta_{i}\right)\\ {\bar{\rho }}_{i,i+1}-R^{T}\left(\theta_{i}\right)\left(p_{i+1}-p_{i}\right) \end{array}\right]$$

The Jacobian matrices of the wheel speed observation error with respect to the robot poses at two different time points are as follows: (42)$$\frac{\partial e_{i,i+1}}{\partial q_{i}}=\left[\begin{array}{cc} 1 & \;0\\ \;R^{T}{\left(p_{i+1}-p_{i}\right)}^{\wedge } & R^{T} \end{array}\right]$$
(43)$$\frac{\partial e_{i,i+1}}{\partial q_{i+1}}=\left[\begin{array}{cc} -1 & 0\\ 0 & -R^{T} \end{array}\right]$$

The Jacobian matrix of the translational component of the error with respect to the rotational component of the i-th robot state is calculated using the right perturbation approach. Consequently, the observation error of the wheel speed pre-integration, its associated covariance, and the Jacobian matrix of the error with respect to the variables to be optimized have been obtained.

### 2.5. Loop Closure Co-Visibility Observation Constraint

In addition to the visual observation constraints and wheel speed pre-integration observation constraints introduced in the previous sections, there is also a loop closure constraint in the robot’s movement. This constraint is important for eliminating cumulative errors that arise during the robot’s state estimation process. When the robot’s pose $v_{1}$ at a certain time forms a loop closure with the pose $v_{2}$ at a previous time, there will be a large number of co-visible landmarks between the two frames. This allows the system to construct a loop closure constraint to correct the robot’s pose.

By taking the current frame, the loop closure frame, and the co-visible landmarks as nodes, and using the observation error of the landmarks in the camera coordinate system as edges, a subgraph optimization problem can be constructed. For instance, consider the observation of the k-th landmark ${\bar{P}}_{k}^{c_{i}}$ from the perspective of the current frame $c_{i}$ (defined as $c_{1}$ for the current frame and $c_{2}$ for the loop closure frame). The measurement error can be expressed as
(44)$$e_{c_{i},k}\left(X_{i}\right)={\bar{P}}_{k}^{c_{i}}-T_{c_{i}w}P_{k}$$

The Jacobian matrix of the measurement error $e_{c_{i},k}$ with respect to the optimization variable *X* is
(45)$$\begin{array}{c} \frac{\partial e_{i,k}}{\partial P_{k}}=-R_{c_{i}w}\\ \frac{\partial e_{i,k}}{\partial \delta \psi }=\left[\begin{array}{cc} -I & \;P_{k}^{\wedge } \end{array}\right] \end{array}$$

Let $G_{k}=\left[\begin{array}{cc} \frac{\partial e_{i,k}}{\partial P_{k}} & \frac{\partial e_{i,k}}{\partial \delta \psi } \end{array}\right]$. By applying a first-order Taylor expansion, Equation (Equation 44) can be expressed as
(46)$$e_{c_{i},k}\left(X\right)≃e_{c_{i},k}\left(X^{*}\right)+G_{i,k}\Delta X$$

Therefore, the error function to be optimized is
(47)$$J\left(X\right)=\frac{1}{2}\sum_{i=1}^{2}\sum_{k=1}^{N}{\left(e_{c_{i},k}\left(X^{*}\right)+G_{i,k}\Delta X\right)}^{T}\Sigma_{k}^{-1}\left(e_{c_{i},k}\left(X^{*}\right)+G_{i,k}\Delta X\right)$$

After the iteration converges, the optimized poses of the two frames can be obtained as poses $T_{c_{1}w}^{*}$ and $T_{c_{2}w}^{*}$.

In the next step, to obtain the covariance of the relative observations, we need to calculate the covariance of each pose individually. Under the assumption of a Gaussian distribution, let the current pose to be optimized and the loop closure pose be variables $X={\left[X_{1},\;X_{2}\right]}^{T}$, respectively, while the shared 3D point is denoted as *P*. The joint probability distribution of (X,P) can be represented as the product of the marginal and conditional probabilities [28], which can be expressed as follows: (48)$$P\left(X,P\right)=P\left(X\right)P\left(P|X\right)\propto \exp\left(\frac{1}{2}{\left[\begin{array}{c} X\\ P \end{array}\right]}^{T}{\left[\begin{array}{cc} \Sigma_{XX} & \;\Sigma_{XP}\\ \Sigma_{PX} & \;\Sigma_{PP} \end{array}\right]}^{-1}\left[\begin{array}{c} X\\ P \end{array}\right]\right)$$

After iterative optimization, the Hessian matrix used for solving the optimization problem in the graph can be obtained. This matrix is denoted as *H*: (49)$$H=\left[\begin{array}{cc} H_{ii} & H_{ik}\\ \;H_{ki} & H_{kk} \end{array}\right]$$

The matrix block $H_{ii}\left(12\times 12\right)$ in the top left corner corresponds to the current frame and loop closure frame, while the matrix block $H_{kk}\left(3k\times 3k\right)$ in the bottom right corner corresponds to the landmark points.

For the likelihood distribution, the information matrix is the expected Hessian matrix of the negative log-likelihood problem [29]. Given the information matrix of the joint distribution, the information matrix of the marginal probability can be derived through decomposition [30]. By marginalizing the landmark points using the Schur complement, the information matrix for the marginal probability P(X) can be obtained as follows: (50)$${H^{\prime }}_{ii}=H_{ii}-\;H_{ik}H_{ik}^{-1}H_{ki}$$

Next, we rewrite ${H^{\prime }}_{ii}$ in the following form: (51)$${H^{\prime }}_{ii}=\left[\begin{array}{cc} {H^{\prime }}_{11} & \;{H^{\prime }}_{12}\\ \;{H^{\prime }}_{21} & \;{H^{\prime }}_{22} \end{array}\right]$$

Assuming the current frame pose and loop closure frame pose are mutually independent, their information matrices correspond to the respective blocks in ${H^{\prime }}_{ii}$. Since the covariance matrix is the inverse of the information matrix, the information matrices for the current frame and the loop closure frame can be obtained as follows: (52)$$\Sigma_{11}^{-1}=\;{H^{\prime }}_{11}$$
(53)$$\Sigma_{22}^{-1}=\;{H^{\prime }}_{22}$$

Consequently, the perturbed poses of the current frame and the loop closure frame can be represented as follows: (54)$${\bar{T}}_{c_{1}w}=\exp\left(\epsilon_{1}^{\wedge }\right)T_{c_{1}w}^{*}$$

In the context of this statement, $\epsilon_{1}\in R^{6},\epsilon_{1}∼N\left(0,\Sigma_{11}\right)$ refers to the left perturbation in small quantities.

Similarly: (55)$${\bar{T}}_{c_{2}w}=T_{c_{2}w}^{*}\exp\left(\epsilon_{2}^{\wedge }\right)$$

In the context of this statement, $\epsilon_{1}\in R^{6},\epsilon_{1}∼N\left(0,\Sigma_{11}\right)$ refers to the right perturbation in small quantities. Both the left and the right perturbations are equivalent; however, we choose the right perturbation in this context for convenience in subsequent calculations.

Therefore, the relative observation between the current frame and the loop closure frame is expressed as follows: (56)$${\bar{T}}_{12}={\bar{T}}_{c_{1}w}*{\bar{T}}_{c_{2}w}^{-1}$$

Given the relative observation perturbation as ϵ, the perturbation mechanism leads to the following equation: (57)$$\exp\left(\epsilon^{\wedge }\right){\bar{T}}_{12}=\exp\left(\epsilon_{1}^{\wedge }\right)T_{c_{1}w}{\left(T_{c_{2}w}\exp\left(\epsilon_{2}^{\wedge }\right)\right)}^{-1}$$

From identity,
(58)$$\exp\left({\left(Tx\right)}^{\wedge }\right)=T\exp\left(x^{\wedge }\right)T^{-1}$$

We obtain
(59)$$\exp\left(\epsilon^{\wedge }\right){\bar{T}}_{12}=\exp\left(\epsilon_{1}^{\wedge }\right)T_{c_{1}w}\exp\left({\left(-T_{c_{1}w}\epsilon_{2}\right)}^{\wedge }\right)T_{c_{1}w}T_{c_{2}w}$$
which is
(60)$$\exp\left(\epsilon^{\wedge }\right)=\exp\left(\epsilon_{1}^{\wedge }\right)\exp\left({\left(-T_{c_{1}w}\epsilon_{2}\right)}^{\wedge }\right)$$

Given the formula ${\epsilon^{\prime }}_{2}=-T_{c_{1}w}\epsilon_{2}$, the covariance of ϵ (to the second-order term only) can be derived using the Baker–Campbell–Hausdorff (BCH) formula: (61)$${\bar{T}}_{12}=\exp\left(\epsilon^{\wedge }\right)T_{12}$$
where $\epsilon ∼N(0,\Sigma_{12})$.

Therefore, the observation error between the loop closure frame and the current frame can be expressed as
(62)$$e_{12}=\log{\left({\bar{T}}_{12}T_{wb_{1}}T_{wb_{2}}^{-1}\right)}^{\vee }$$

Similar to Equation (Equation 40), the transformation can be expressed in terms of the rotation and translation components on SE(2) as follows:(63)$$\left[\begin{array}{c} e_{12}^{\theta }\\ e_{12}^{\rho } \end{array}\right]=\left[\begin{array}{c} {\bar{\theta }}_{12}-\left(\theta_{1}-\theta_{2}\right)\\ {\bar{\rho }}_{12}-R^{T}\left(\theta_{1}\right)\left(p_{2}-p_{1}\right) \end{array}\right]$$

The corresponding Jacobian matrix is similar to that in Equations (42) and (43), so it will not be repeated here. Thus, the loop closure covisual observation error, covariance matrix, and Jacobian matrix relative to the optimization variables have been obtained.

### 2.6. Formulation of the Maximum A Posteriori Estimation Problem

The essence of robot state estimation is maximum a posteriori estimation given known observations; when ignoring unknown prior information, this can be converted to maximum likelihood estimation. Under the assumption that the probability density function of the state variables follows a Gaussian distribution, the problem can be further transformed into a least squares estimation problem through negative logarithmic transformation. Thus, by treating the mobile robot pose and map points as state variables $\chi =(T_{wb_{1}},\cdots ,T_{wb_{n}},P_{1}^{w},\cdots ,P_{m}^{w})$, and based on the aforementioned reprojection error Equation (Equation 25), wheel speed pre-integration error Equation (Equation 40), and loop closure co-visibility error Equation (Equation 62), a nonlinear error function can be constructed as follows: (64)$$F\left(\chi \right)=\sum_{in,jm}e_{ij}^{T}\Sigma_{ij}^{-1}e_{ij}+\sum_{in-1}e_{i,i+1}^{T}\Sigma_{i,i+1}^{-1}e_{i,i+1}+\lambda \sum_{ik}e_{12}^{T}\Sigma_{12}^{-1}e_{12}$$

In the equation, λ takes a value of 0 or 1, where it is set to 1 when loop closure and global optimization are needed.

The optimal state vector $\chi^{*}$ is obtained by minimizing the objective function F(χ): (65)$$\chi^{*}=\underset{\chi }{\arg\min}\;F\left(\chi \right)$$

The error function can be linearized at the operating point χ using the second-order Taylor expansion, which yields
(66)$$F\left(\chi +\delta \chi \right)\approx F\left(\chi \right)+\;J\delta \chi +\frac{1}{2}\delta \chi^{T}H\delta \chi$$

To minimize the objective function, the function is differentiated with respect to δχ and set to zero. This results in the optimal perturbation value of the state vector, which is the solution to the following linear system: (67)$$H\delta \chi^{*}=-J$$

The optimal solution for the state variables is the best estimate of the pose of the wheeled mobile robot and the observed 3D map points: (68)$$\chi^{*}\leftarrow \;\chi^{*}⊞\delta \chi^{*}$$

In this context, ⊞ represents the generalized summation symbol.

## 3. Experimental Comparison and Analysis

To verify the state estimation performance of the proposed Wheel Speed Visual SLAM (W-VSLAM) method—henceforth referred to as W-VSLAM in the text—for planar robot motion in an indoor environment without GPS, a cross-comparison with the latest visual SLAM algorithms in the field of visual methods was conducted. This comparison includes qualitative visual analysis and quantitative analysis based on general evaluation metrics, aiming to comprehensively validate the superiority of the proposed method. This section begins by detailing the experimental dataset, including the necessary parameters for different forms of data, and introduces two commonly used trajectory accuracy evaluation metrics in the SLAM field. Next, it provides an in-depth explanation of two sets of comparison experiments to validate the superiority of the W-VSLAM method. Finally, three groups of accuracy evaluation experiments are conducted to verify the effectiveness of the proposed W-VSLAM method.

### 3.1. Data Collection

Given that existing public datasets do not include indoor SLAM data with robot speed information or camera-encoder extrinsic parameters, a dataset was collected using a self-developed mobile robot platform in an indoor environment of an academic building. To facilitate comparison with open-source visual algorithms, the camera was mounted on top of the robot body in an upward-facing orientation (front-facing camera orientation rendered the latest visual SLAM algorithms completely ineffective). The encoders, coaxial with the drive motors, provided feedback on the wheel rotation speed of the robot platform at 1024 lines. The remotely controlled robot platform navigated an indoor environment to form a looped trajectory. During this motion, data from a 2D lidar, mounted on the front of the robot and operating at a frequency of 10 Hz, were recorded. In addition, gray-scale images were captured at a frequency of 25 Hz with a resolution of 752 × 480 pixels. Wheel speed data, calculated at the base layer, were collected at a frequency of 100 Hz and measured in RPM (revolutions per minute). The coefficients of the radial–tangential distortion model for the camera are shown in Table 1. The intrinsic parameters matrix *K* for the pinhole camera model are as follows: (69)$$K=\left[\begin{array}{ccc} 366.9826923 & 0 & 361.35791565\\ 0 & 366.7935063 & 246.71152313\\ 0 & 0 & 1 \end{array}\right]$$

### 3.2. Quantitative Evaluation Metrics

To assess the performance of the proposed vision algorithm, it must be compared with the ground truth trajectory, which is typically acquired using more accurate sensors. Common trajectory evaluation metrics include Absolute Trajectory Error (ATE) and Relative Pose Error (RPE). Let $T_{k}$ represent the pose of the trajectory being evaluated at time *k*, and let $T_{gt,k}$ represent the pose of the ground truth trajectory at the same time: (70)$$\mathrm{AT}E_{k}=\log{\left(T_{gt,k}^{-1}T_{k}\right)}^{\vee }$$

In the equation, $\log{\left(\cdot \right)}^{\vee }$ represents the operator that maps a transformation matrix (Lie group) to a vector (Lie algebra).

After obtaining the *N* trajectory errors for the two trajectories, statistical calculations can be performed to obtain common SLAM trajectory evaluation metrics, such as root mean square error (RMSE): (71)$$\mathrm{AT}E_{\mathrm{RMSE}}=\sqrt{\frac{1}{N}\sum_{k=1}^{N}{\left|\left|\log{\left(T_{gt,k}^{-1}T_{k}\right)}^{\vee }\right|\right|}^{2}}$$

After obtaining the absolute trajectory error, it is also necessary to evaluate the relative trajectory error. This can be conducted by considering the trajectory from time *k* to time k+Δt and calculating the relative trajectory error as follows: (72)$$\mathrm{RP}E_{k}=\log{\left({\left(T_{gt,k}^{-1}T_{gt,k+\Delta t}\right)}^{-1}\left(T_{k}^{-1}T_{k+\Delta t}\right)\right)}^{\vee }$$

Similarly, the root mean square error (RMSE) for the relative trajectory error can be calculated as follows: (73)$$\mathrm{RP}E_{\mathrm{RMSE}}=\sqrt{\frac{1}{N}\sum_{k=1}^{N}{\left|\left|\log{\left({\left(T_{gt,k}^{-1}T_{gt,k+\Delta t}\right)}^{-1}\left(T_{k}^{-1}T_{k+\Delta t}\right)\right)}^{\vee }\right|\right|}^{2}}$$

In addition to evaluating the trajectory as a whole, one can decompose the pose into translation and rotation components and evaluate each separately. The calculation method is similar to the overall evaluation but involves extracting the corresponding translation or rotation components and calculating the differences. In the subsequent experiments, this separate evaluation approach is used, calculating the maximum error, minimum error, and root mean square error (RMSE) for each component. This allows for a detailed comparison of the estimation performance of the algorithm.

Due to the indoor environment without GPS, the robot’s true trajectory is obtained using the common laser HectorSLAM method with a 2D lidar. Then, using the ATE and RPE evaluation metrics discussed earlier, the proposed method, the latest open-source visual algorithm ORBSLAM3, and the wheel odometry trajectories are compared in terms of error accuracy relative to the true trajectory. This allows for a cross-comparison of the methods.

It should be noted that the other well-known open-source algorithms, VINS-Mono and VINS-Fusion, are not included in the comparison because the accuracy of the VINS algorithms relies significantly on precise initialization. For wheeled mobile robots in planar motion, it is difficult to fully excite the accelerometer during initialization, which leads to the divergence of the estimated robot trajectory. This renders the algorithm ineffective and not suitable for comparison. Therefore, wheel odometry trajectories are used as a substitute.

### 3.3. Indoor Comparative Experimental Results and Analysis

To validate the superiority of the proposed method for the state estimation of planar motion robots in indoor environments, a cross-comparison was conducted in the visual SLAM field using two sets of experiments. The comparison dataset was obtained by manually controlling a robot to follow a square trajectory in an indoor environment, as depicted in Figure 5. During the movement, there were lighting changes, and at the corners, there were pure rotations of 90 degrees. The first set of data involved running two laps at a speed of 0.08 m/s, while the second set of data consisted of one lap at a speed of 0.04 m/s (the two sets of data were not meant to be controlled for; the number of laps was chosen randomly for ease of operation, so one set involved one lap, and the other involved two laps). Figure 6 shows the feature points tracked during the algorithm’s execution. Matched feature points in the reference frame are indicated by red circles, while unmatched points are shown in blue circles. Matched feature points in the current frame are represented by green circles, while unmatched points are represented by red circles. The results of running the proposed SLAM algorithm are shown in Figure 7 and Figure 8. Black points represent landmark points, black boxes represent keyframes, blue lines represent the co-visibility between keyframes, and red represents the current orientation and landmark points.

The trajectory obtained by Laser HectorSLAM is considered the ground truth. The trajectories of the proposed W-VSLAM method, the latest visual open-source algorithm ORBSLAM3 keyframe trajectory, and the robot’s odometry trajectory were compared. The trajectory comparisons for the two sets of experiments are shown in Figure 9 and Figure 10. Given the inconsistency in time frequencies across the different trajectories, the maximum time error is set at 0.01 s. Additionally, due to the external parameter transformation between different sensors, the results were aligned using the visualization tool.

In Figure 9, the 2DLaser (ref) represents the reference trajectory obtained by 2DLaser and is shown as a dashed line. Ours represents the proposed W-VSLAM method and is depicted as a blue solid line. Odom represents the trajectory obtained from odometry and is depicted as a green solid line. ORBSLAM3 represents the latest open-source algorithm and is depicted as a solid red line. In Figure 10, the 2DLaser (ref) represents the reference trajectory obtained by Laser SLAM and is shown as a dashed line. Odom represents the trajectory obtained from odometry and is depicted as a solid blue line. Ours represents the proposed W-VSLAM method and is depicted as a solid green line. ORBSLAM3 represents the latest open-source algorithm and is depicted as a solid red line. From the two figures, it is visually apparent that the trajectory obtained using the ORBSLAM3 algorithm has significant errors in the *z*-axis direction, whereas the proposed visual algorithm aligns more closely with the reference trajectory.

More specifically, in Experiment One, the comparison of trajectories obtained by the four methods in terms of translation is illustrated in Figure 11. In the *x* and *y* axes, the odometry trajectory experiences drift due to accumulated error from integration. Similarly, the ORBSLAM3 algorithm requires initialization, resulting in a missing segment at the beginning and also showing accumulated error, though less than the odometry error. In contrast, the proposed W-VSLAM method aligns more closely with the reference trajectory, displaying a smaller error. In the z-axis direction, the proposed algorithm uses a 3-degree-of-freedom representation for the robot’s pose, ensuring an accurate estimation of planar motion. Meanwhile, the ORBSLAM3 algorithm uses a six-degree-of-freedom parameterization for the robot’s pose during planar motion, leading to erroneous estimation. This algorithm also exhibits significant errors in the pitch and roll directions, which are not presented here. The results of Experiment Two are similar and are not included.

After the qualitative analysis presented above, a quantitative analysis of the trajectory accuracy of the various algorithms in the two comparison experiments was conducted using two evaluation metrics: absolute trajectory error (ATE) and relative pose error (RPE). The results for Experiment One are shown in Table 2 and Table 3, while the results for Experiment Two are presented in Table 4 and Table 5. For relative pose error (RPE), considering the variation in time frequencies of the different trajectories, the relative error was calculated over a relative increment of 1 m in the translation direction and a relative increment of 1 degree in the rotation estimation.

Absolute trajectory error reflects the algorithm’s accuracy over the entire trajectory. In Experiment One, as shown in Table 2, the proposed algorithm achieved a root mean square error of 0.135999 m in the translation direction, which is significantly better than the other two algorithms. The root mean square error in rotation estimation was 0.024885 rad, also outperforming the other two algorithms.

In Experiment Two, as shown in Table 4, the root mean square error in translation estimation was 0.160804 m, while the root mean square error in rotation estimation was 0.101190 rad. In horizontal comparisons, the proposed algorithm outperformed the other two algorithms, demonstrating its ability to integrate two relatively imprecise sensor data streams to achieve more accurate pose estimation and validate the effectiveness of the algorithm.

Moreover, a vertical comparison of Table 2 and Table 4 reveals that the translation estimation in Experiment One is superior to that in Experiment Two, as is the rotation estimation. This is likely because the trajectory in Experiment One had more loop closures, resulting in more accurate pose estimation for the robot. This finding further supports the effectiveness of the proposed algorithm. Relative trajectory error reflects the algorithm’s performance on local trajectories.

In Experiment One, as shown in Table 3, the proposed algorithm achieved a root mean square error of 0.035044 m in translation estimation and 0.013080 rad in rotation estimation. Experiment Two showed similar results, with a root mean square error of 0.094710 m in translation estimation and 0.010742 rad in rotation estimation. In horizontal comparisons, these results are slightly below the best performance, but the differences are minimal. It is notable that while the wheel odometry method accumulates errors quickly due to high-frequency integration, resulting in inaccurate global trajectory estimation, it is relatively accurate in local relative pose estimation. The proposed visual SLAM algorithm, which integrates wheel odometry, leverages this aspect. By using pre-integration to obtain accurate inter-camera frame estimations as constraints, and by shortening the wheel odometry integration interval to reduce cumulative errors, the algorithm improves the relative pose estimation performance of the camera. This ensures that the proposed algorithm can achieve more accurate global pose estimation.

In summary, for planar motion robot state estimation under significant lighting variations, the proposed method demonstrates superior performance in both global and local accuracy compared to the latest visual algorithms in the field. When compared to wheel odometry, the proposed method also exhibits superior performance globally, while achieving nearly comparable, albeit slightly lower, performance at the local level.

### 3.4. Results and Analysis of Multiple Indoor Experiments

Additionally, three experimental sets were conducted multiple times to validate the robustness and perceptual performance of the proposed W-VSLAM method, mitigating potential environmental factors. The comparative results with the latest visual SLAM algorithms have been discussed in the previous section and will not be repeated here. Instead, the trajectories provided by laser HectorSLAM are used as reference true values to evaluate the perceptual accuracy of the proposed visual SLAM method. The results of the robot’s environmental perception across different motion trajectories, as well as comparisons with the reference trajectory, are shown in Figure 12, Figure 13, Figure 14, Figure 15, Figure 16, Figure 17 and Figure 18. The RMSE accuracy of the W-VSLAM method across different trajectories is presented in Table 6. In the first dataset, the robot initially follows a small square trajectory and then expands its movement range. In the second dataset, the robot traverses two adjacent closed square trajectories. The third dataset involves a back-and-forth motion pattern. The first two datasets mainly highlight the impact of lighting changes during the robot’s movement on the perception algorithm, while the third emphasizes the effect of a 360-degree pure rotation at the midpoint of a straight trajectory.

The results from the three different experimental sets indicate that the proposed W-VSLAM perception algorithm can robustly perceive robot poses and map points even under conditions of strong lighting changes or 360-degree pure rotational motion. The trajectory localization accuracy ranges from 5 cm to 17 cm, aligning with the current perception levels of visual SLAM methods and further verifying the effectiveness of the proposed approach.

### 3.5. The Results and Analysis of the Indoor Long Corridor Environment Experiment

The indoor long corridor environment has always posed challenges for laser SLAM perception. Due to the limited range or angle of sensors, the information perceived by the sensors does not significantly change during movement, leading to inaccurate state estimation. In this experiment, a robot was controlled to navigate through a long corridor within an indoor environment, returning through the corridor to form a closed-loop trajectory. Throughout the movement, significant changes in illumination intensity were observed. The results of the proposed W-VSLAM method in this long corridor environment are shown in Figure 15.

In the aforementioned corridor environment, we operated HectorSLAM using a 2D LiDAR with a scan range parameter set to 5 m. The resulting trajectory was compared with the trajectories obtained using the proposed algorithm and ORBSLAM3. The visual comparisons of the trajectories are illustrated in Figure 16 and Figure 17. Based on the conclusions from the quantitative accuracy comparison in the previous subsection, the trajectory from the wheel speed odometer was neither included in the comparison nor reported. Observing the trajectory obtained by HectorSLAM with a 5 m detection range, it is evident that when the sensor’s detection distance is insufficient to cover the corridor length, the trajectory generated by HectorSLAM becomes distorted. This distortion occurs because the point cloud observations within the detection range do not change as the robot moves through the corridor, leading to a lack of translational estimation (in the y-axis direction in the world coordinate system). In contrast, the proposed algorithm can still accurately perceive the environment in the long corridor, demonstrating the advantages of visual SLAM methods in specific settings. Additionally, as seen in the translational component comparison in Figure 17, the trajectory obtained by ORBSLAM3 is generally accurate in the r-axis direction (perpendicular to the corridor direction), except at the end of the trajectory. The proposed algorithm’s trajectory closely aligns with the reference trajectory, except for a noticeable deviation at the trajectory’s end due to visual odometry estimation errors. The trajectory generated by the proposed method is closer to the actual path, further showcasing its advantages in environments where LiDAR fails.

Although the translational estimation of HectorSLAM’s reference trajectory is inaccurate and thus cannot be analyzed in detail, its rotational estimation can still be qualitatively assessed. The absolute and relative trajectory accuracies for rotational estimation, as achieved by the proposed algorithm compared to the reference trajectory, are shown in Figure 18 and Figure 19, respectively. In addition to the maximum and minimum errors illustrated in the figures, the root mean square error (RMSE) of the absolute rotational error is 0.0726080 rad, and the RMSE of the relative rotational error is 0.022059 rad, which is consistent with the results from the previous subsection. Overall, the comparison with the HectorSLAM laser algorithm demonstrates that the proposed visual SLAM algorithm can achieve more accurate perception in indoor long corridors or diffuse reflection environments where 2D LiDAR fails.

## 4. Conclusions

To address the estimation inaccuracies in indoor environments without GPS signals for ground mobile robots, caused by excess pose degrees of freedom and the lack of excitation in accelerometers, a visual SLAM perception method that integrates wheel odometry information is proposed. The method parameterizes the robot pose on SE(2) and the camera pose on SE(3). It then derives visual constraint residuals and Jacobian matrices for reprojection observations using the camera projection model and uses the pre-integration concept to derive pose-constraint residuals and Jacobian matrices from wheel odometry pre-integration quantities. Loop closure observation constraints are derived using marginalization theory to calculate relative pose residuals and Jacobians. These are used to solve the nonlinear optimization problem and obtain the optimal poses and landmark points for the ground mobile robot. Comparisons with the ORBSLAM3 algorithm and wheel odometry in experimental tests indicate that the proposed visual SLAM method integrating wheel odometry provides higher perception accuracy in planar motion estimation, particularly in indoor environments with significant lighting variations or pure rotational motion. The method’s superiority is demonstrated with accuracy in a range of 5–17 cm across multiple experiments, further confirming its effectiveness. It is worth noting that the proposed visual SLAM perception method with integrated wheel odometry is not limited to wheeled mobile platforms; it can also be applied to tracked mobile platforms.

## Figures and Tables

**Figure 1 sensors-24-05662-f001:**
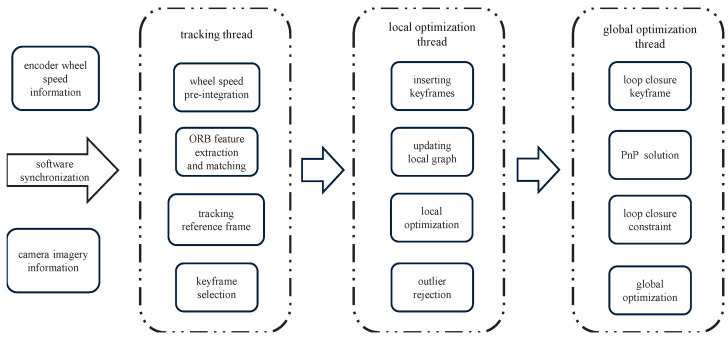
Block diagram of mobile robot visual SLAM with integrated wheel speed.

**Figure 2 sensors-24-05662-f002:**
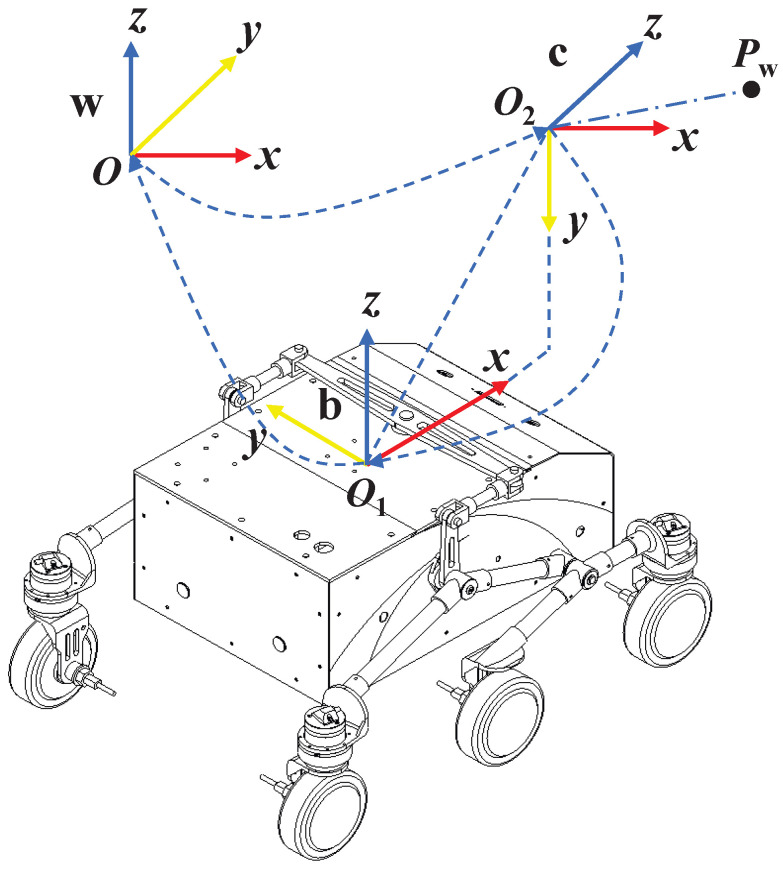
Schematic diagram of mobile robot coordinate system.

**Figure 3 sensors-24-05662-f003:**
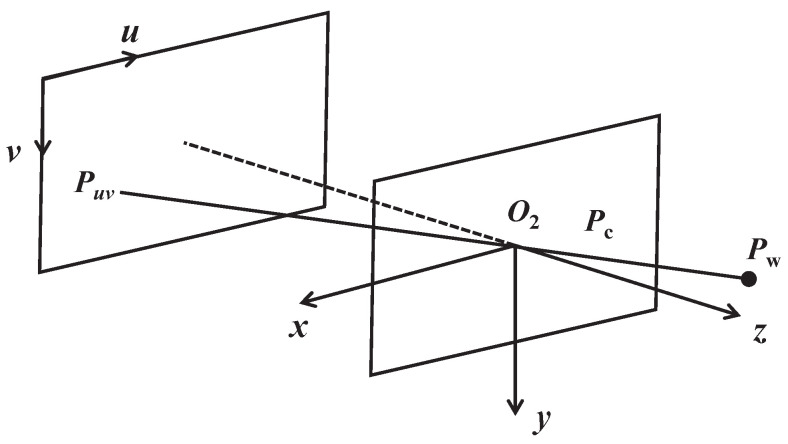
Camera projection model.

**Figure 4 sensors-24-05662-f004:**
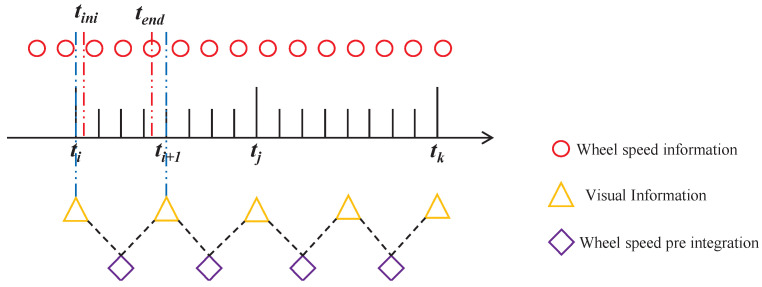
Schematic diagram of wheel speed information pre-integration.

**Figure 5 sensors-24-05662-f005:**
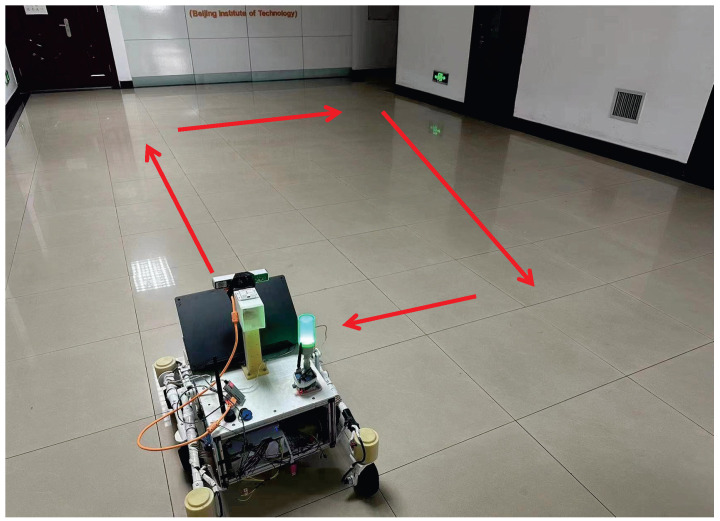
Schematic diagram of square movement in the comparative experiment.

**Figure 6 sensors-24-05662-f006:**
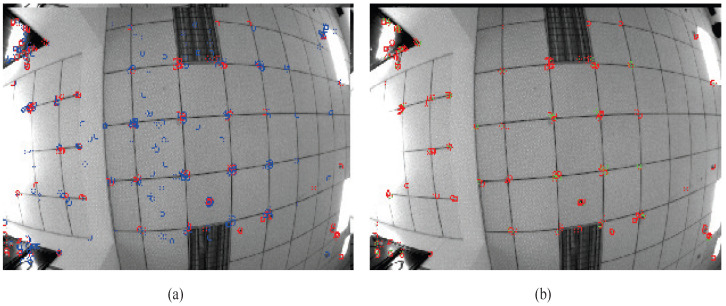
Reference keyframe and current frame in previous tracking, (**a**) reference keyframe of previous tracking and (**b**) current frame of previous tracking.

**Figure 7 sensors-24-05662-f007:**
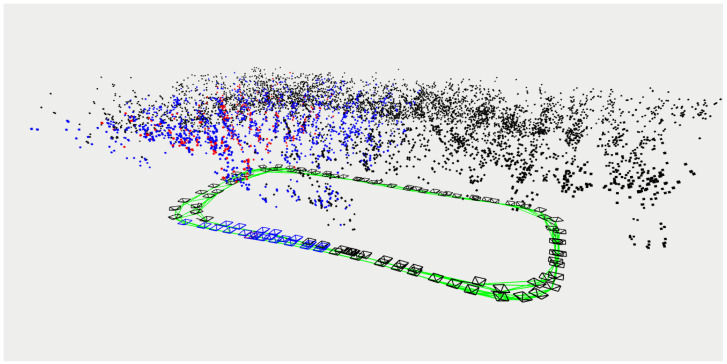
Comparative Experiment One: robot poses and environmental map points obtained by W-VSLAM.

**Figure 8 sensors-24-05662-f008:**
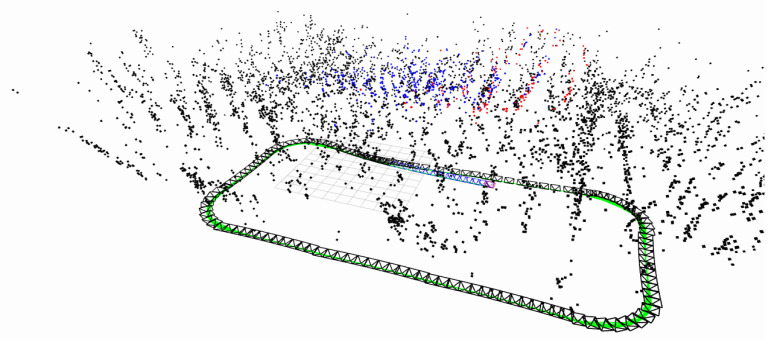
Comparative Experiment Two: robot poses and environmental map points obtained by W-VSLAM.

**Figure 9 sensors-24-05662-f009:**
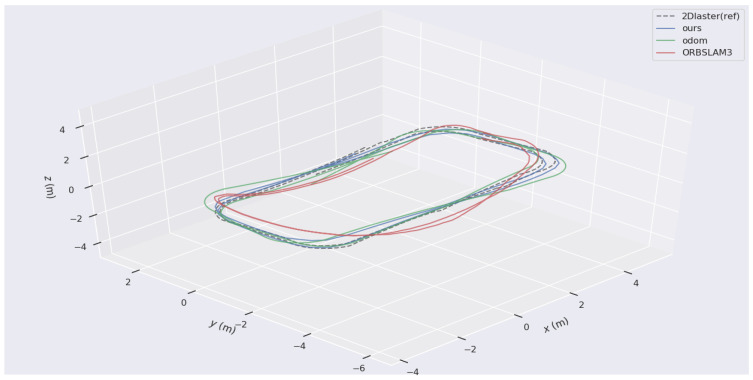
Comparative Experiment One: trajectory comparison chart of different algorithms.

**Figure 10 sensors-24-05662-f010:**
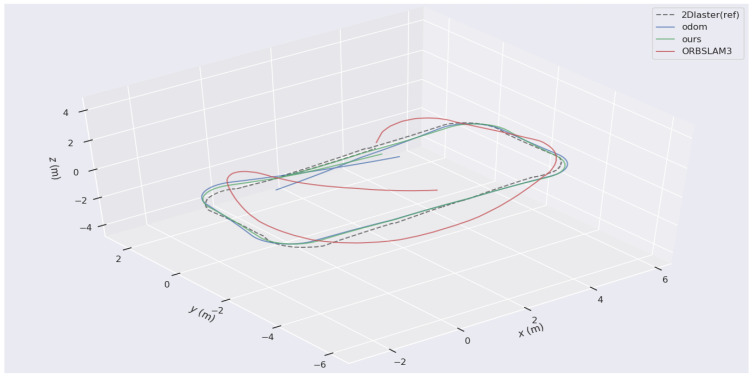
Comparative Experiment Two: trajectory comparison chart of different algorithms.

**Figure 11 sensors-24-05662-f011:**
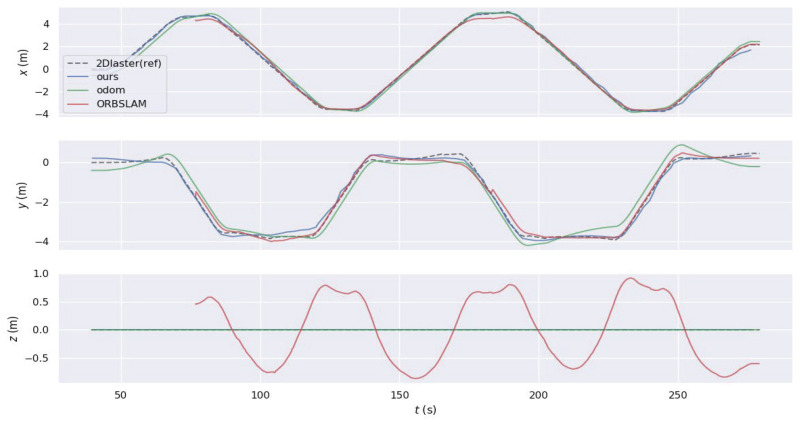
Comparative test one: translational component comparison chart of trajectories from different algorithms.

**Figure 12 sensors-24-05662-f012:**
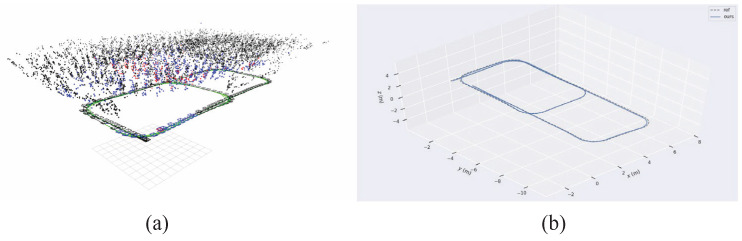
(**a**) Perception results of robot poses and map points in Experiment One. (**b**) Comparison between Experiment One and reference trajectory.

**Figure 13 sensors-24-05662-f013:**
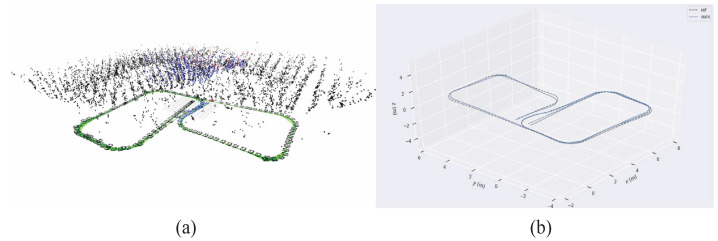
(**a**) Perception results of robot poses and map points in Experiment Two. (**b**) Comparison between Experiment Two and reference trajectory.

**Figure 14 sensors-24-05662-f014:**
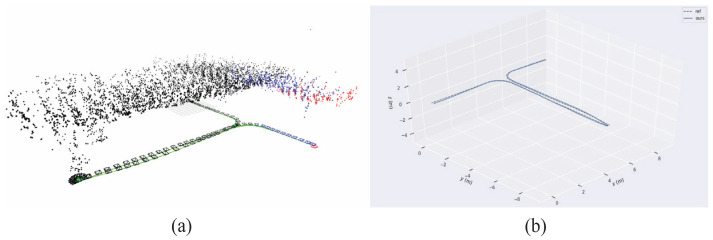
(**a**) Perception results of robot poses and map points in Experiment Three. (**b**) Comparison between Experiment Three and reference trajectory.

**Figure 15 sensors-24-05662-f015:**
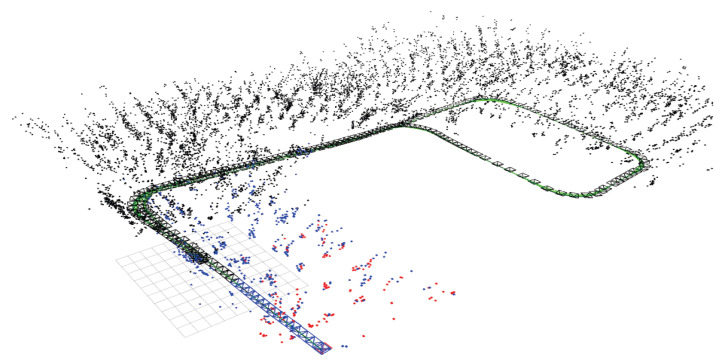
Indoor long corridor environment trajectory; Rivz result diagram.

**Figure 16 sensors-24-05662-f016:**
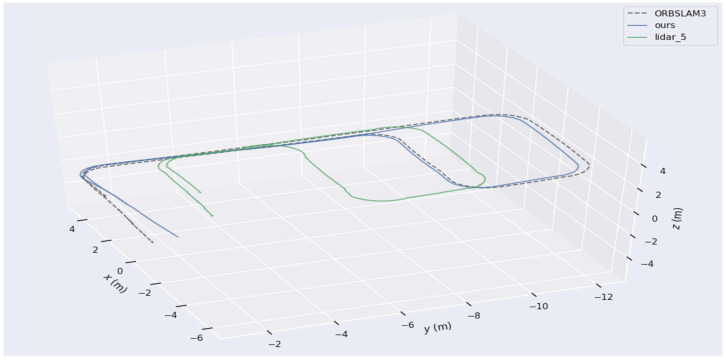
Comparison chart of trajectories in the indoor long corridor environment.

**Figure 17 sensors-24-05662-f017:**
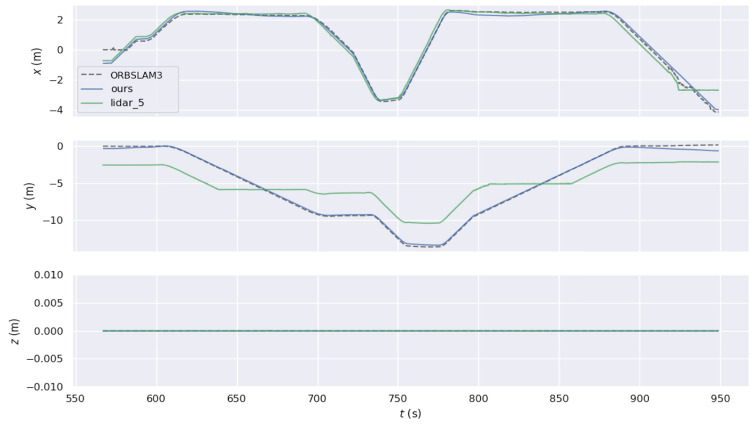
Comparison chart of translational components of trajectories in the indoor long corridor environment.

**Figure 18 sensors-24-05662-f018:**
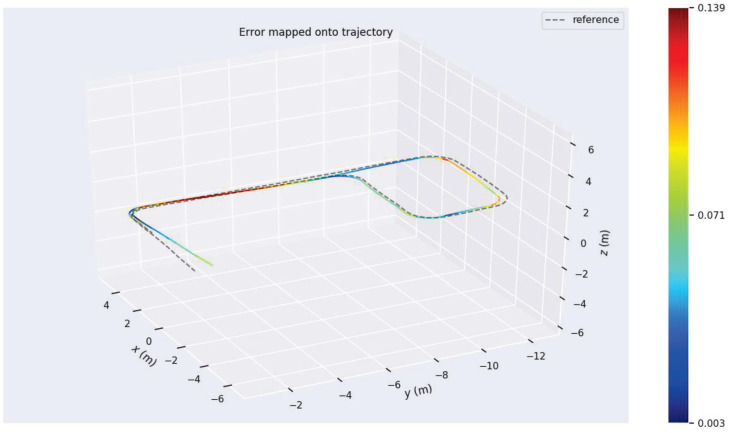
Absolute accuracy of rotational estimation in the indoor long corridor environment.

**Figure 19 sensors-24-05662-f019:**
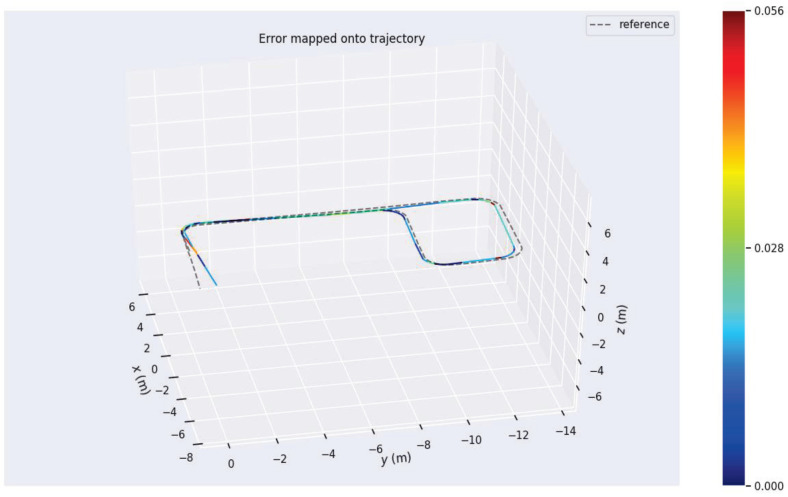
Relative accuracy of rotational estimation in the indoor long corridor environment (with a 1° increment).

**Table 1 sensors-24-05662-t001:** Performance index data of different segmentation algorithms.

Parameters	Radial Distortion $k_{1}$	Radial Distortion $k_{2}$	Tangential Distortion $p_{1}$	Tangential Distortion $p_{2}$
Value	−0.02209723	−0.00543498	0.00179311	−0.00069370

**Table 2 sensors-24-05662-t002:** Absolute trajectory error (ATE) estimates of different algorithms for two loop trajectories.

Error Type	ORBSLAM3 Algorithm	Wheel Odometer	The Proposed Algorithm
Maximum Translational Error (m)	0.996424	0.663076	0.135999
Minimum Translational Error (m)	0.053682	0.191564	0.005725
Translational Root Mean Square Error (m)	0.649618	0.411756	0.070133
Maximum Rotational Error (rad)	1.798408	0.249882	0.073018
Minimum Translational Error (rad)	1.499492	0.000036	0.000234
Rotational Root Mean Square Error (rad)	1.674251	0.138140	0.024885

**Table 3 sensors-24-05662-t003:** Relative trajectory error estimates of different algorithms for two loop trajectories.

Error Type	ORBSLAM3 Algorithm	Wheel Odometer	The Proposed Algorithm
Maximum Translational Error (m)	2.258283	0.072512	0.083033
Minimum Translational Error (m)	1.679300	0.004918	0.005750
Translational Root Mean Square Error (m)	1.934542	0.030060	0.035044
Maximum Rotational Error (rad)	0.267859	0.015824	0.042052
Minimum Translational Error (rad)	0.004864	0.000011	0.000113
Rotational Root Mean Square Error (rad)	0.080078	0.002707	0.013080

**Table 4 sensors-24-05662-t004:** Absolute trajectory error (ATE) estimates of different algorithms for one loop trajectory.

Error Type	ORBSLAM3 Algorithm	Wheel Odometer	The Proposed Algorithm
Maximum Translational Error (m)	2.448048	0.877545	0.265917
Minimum Translational Error (m)	0.270060	0.130613	0.008439
Translational Root Mean Square Error (m)	1.263682	0.384537	0.160804
Maximum Rotational Error (rad)	2.242332	0.288801	0.255336
Minimum Translational Error (rad)	1.169881	0.001404	0.001039
Rotational Root Mean Square Error (rad)	1.844953	0.157194	0.101190

**Table 5 sensors-24-05662-t005:** Relative trajectory error (RPE) estimates of different algorithms for one loop trajectory.

Error Type	ORBSLAM3 Algorithm	Wheel Odometer	The Proposed Algorithm
Maximum Translational Error (m)	1.798007	0.100730	0.120928
Minimum Translational Error (m)	1.286840	0.064568	0.066626
Translational Root Mean Square Error (m)	1.445999	0.083673	0.094710
Maximum Rotational Error (rad)	0.375543	0.026001	0.037211
Minimum Translational Error (rad)	0.007266	0.000009	0.000061
Rotational Root Mean Square Error (rad)	0.069250	0.001821	0.010742

**Table 6 sensors-24-05662-t006:** Results of trajectory accuracy evaluation from multiple experiments.

Category	Experiment 1	Experiment 2	Experiment 3
Absolute Translational RMSE (m)	0.069569	0.174467	0.083839
Absolute Rotation RMSE (rad)	0.079908	0.103241	0.096109
Relative Translation RMSE (m)	0.110006	0.106630	0.107273
Relative Rotation RMSE (rad)	0.012823	0.015159	0.025358

## Data Availability

Data are contained within the article.

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
