# Peer review of "W-VSLAM: A Visual Mapping Algorithm for Indoor Inspection Robots"

_sensors, 2024, doi:10.3390/s24175662_

Round 1

Reviewer 1 Report

Comments and Suggestions for Authors

(1)  Check the writing norms throughout the text; for example, line 185 lacks a corresponding ")".

(2)  Check the title titles again; for example, subsections 2.3 and 2.4 have title names that overlap.

(3)  In practical scenarios, mobile robots often have forward-facing cameras. Therefore, it is recommended to account for the algorithm's performance under this specific configuration.

(4)  The paper by Zheng et al., titled 'Visual-odometric localization and map for ground vehicles using SE(2)-XYZ constraints,' which is cited in the current paper at line 63, also develops a visual SLAM system that incorporates wheel speedometer and camera data. Furthermore, it constructs SE(2)-based visual observation constraints and pre-integration constraints. How does this paper distinguish itself from Zheng et al.'s work in terms of similarities and differences? It is advised to clearly delineate the innovations introduced in this paper."

(5)  For what reasons did the paper not include experimental comparisons with the current wheel SLAM systems based on the SE(3) framework?

Reviewer 2 Report

Comments and Suggestions for Authors

In this work, the authors propose a SLAM method that integrates visual information with encoder-based wheel odometer information. They formulate the pose of a ground robot in the SE(2) space to eliminate redundant pose degrees of freedom and accelerometer drift during planar motion of mobile robots.

This work has a few serious weaknesses. First, its novelty is limited. It is not new to solve the visual SLAM problem specifically for ground robots using wheel odometer information. For example, 10.1109/ICRA.2017.7989603 and 10.1109/ACCESS.2019.2930201 have made thorough explorations. The authors need to provide evidences that this work is significantly different from these work in order to be published.

Second, the experiments are too limited to prove the proposed method effective. The proposed work and baseline methods are only evaluated on a very limited number of data sequences from the authors' custom dataset. Ground truth is generated from LiDAR based SLAM which is also subject to error. It is highly recommended to move evaluation to public datasets such as

KAIST urban dataset https://sites.google.com/view/complex-urban-dataset
 and nuscenes dataset www.nuscenes.org

 which all include wheel odometer data

Comments on the Quality of English Language

2.4. Visual observation constraint has a wrong section name?

inline notation is not in right format: line 225, v and w
